# Chondroitin Sulfate-Tyramine-Based Hydrogels for Cartilage Tissue Repair

**DOI:** 10.3390/ijms24043451

**Published:** 2023-02-09

**Authors:** Ilona Uzieliene, Daiva Bironaite, Jolita Pachaleva, Edvardas Bagdonas, Arkadij Sobolev, Wei-Bor Tsai, Giedrius Kvedaras, Eiva Bernotiene

**Affiliations:** 1Department of Regenerative Medicine, State Research Institute Centre for Innovative Medicine, LT-08406 Vilnius, Lithuania; 2Latvian Institute of Organic Synthesis, LV-1006 Riga, Latvia; 3Department of Chemical Engineering, National Taiwan University, Taipei 104, Taiwan; 4Clinic of Rheumatology, Orthopaedics Traumatology and Reconstructive Surgery, Institute of Clinical Medicine, Faculty of Medicine, Vilnius University, LT-03101 Vilnius, Lithuania

**Keywords:** bone marrow mesenchymal stem cells, chondroitin sulfate tyramine hydrogels, cartilage oligomeric matrix protein, mechanical compression/load, biointegration, cartilage regeneration

## Abstract

The degradation of cartilage, due to trauma, mechanical load or diseases, results in abundant loss of extracellular matrix (ECM) integrity and development of osteoarthritis (OA). Chondroitin sulfate (CS) is a member of the highly sulfated glycosaminoglycans (GAGs) and a primary component of cartilage tissue ECM. In this study, we aimed to investigate the effect of mechanical load on the chondrogenic differentiation of bone marrow mesenchymal stem cells (BM-MCSs) encapsulated into CS-tyramine-gelatin (CS-Tyr/Gel) hydrogel in order to evaluate the suitability of this composite for OA cartilage regeneration studies in vitro. The CS-Tyr/Gel/BM-MSCs composite showed excellent biointegration on cartilage explants. The applied mild mechanical load stimulated the chondrogenic differentiation of BM-MSCs in CS-Tyr/Gel hydrogel (immunohistochemical collagen II staining). However, the stronger mechanical load had a negative effect on the human OA cartilage explants evaluated by the higher release of ECM components, such as the cartilage oligomeric matrix protein (COMP) and GAGs, compared to the not-compressed explants. Finally, the application of the CS-Tyr/Gel/BM-MSCs composite on the top of the OA cartilage explants decreased the release of COMP and GAGs from the cartilage explants. Data suggest that the CS-Tyr/Gel/BM-MSCs composite can protect the OA cartilage explants from the damaging effects of external mechanical stimuli. Therefore, it can be used for investigation of OA cartilage regenerative potential and mechanisms under the mechanical load in vitro with further perspectives of therapeutic application in vivo.

## 1. Introduction

Cartilage is known as an avascular and aneural tissue, consisting of chondrocytes located in specific niches called lacuna and various components of the extracellular matrix (ECM) [1]. The avascular nature of cartilage, low transportation of nutrients and waste products by diffusion and convection through the ECM limits the self-regenerative properties of cartilage [2,3]. The low metabolic activity together with the various types of mechanical joint damages, chronic diseases and/or age-related cartilage impairments usually cause joint degeneration, leading to the development of OA [4]. The early diagnosis of cartilage lesions is challenging since primary pain symptoms occurring from the bone-to-bone rubbing are observed only in the advanced stage of cartilage tissue degeneration [1]. Currently, the treatment of joint impairments is usually based on the reduction of pain, articular physiotherapy, intraarticular injections of hyaluronic acid (HA) or total joint replacement at the end stage of cartilage degeneration [4]. Therefore, the main challenge of articular cartilage therapy is to search for new and more effective means of positively affecting cartilage regeneration.

Recent progress in cartilage regenerative field showed potential of cell-based therapies. Thus far, the Food and Drug Administration has approved the only cellular-based therapy—an implantation of autologous chondrocytes [5]. However, the application of autologous chondrocytes has some limitations, i.e., an invasive procedure for cell obtaining, low cell number, survival and regenerative potential of autologous chondrocytes particularly derived from the OA cartilage. Therefore, other cell sources, such as BM-MCSs, adipo tissue-derived MSCs and others are currently under phase I/II clinical trials for the OA treatment and/or cartilage regenerative purposes [5,6]. From all MSCs types, BM-MCSs are the most widely used for cartilage tissue engineering due to their great chondrogenic differentiation capability and self-renewal capacities [7]. Despite the high potential of BM-MSCs-based OA articular cartilage treatment, there are still numerous problems, such as the poor survival and quick release of the intraarticularly injected cell under the mechanical load [8]. Therefore, the application of injectable hydrogels with encapsulated BM-MSCs, or other types of cells, could be a solution in helping to overcome the mentioned problems [2].

The delivery of cells or various bioactive compounds to the joint using injectable polymeric hydrogels is a promising tool for OA cartilage regeneration since hydrogels can reach defected surfaces via minimally invasive and less painful procedure without a need to open the joint [9]. In addition, the hydrogels used for cartilage regeneration must possess biocompatibility and resistance to the mechanical loads. Therefore, the hydrogels composed from natural cartilage ECM components, such as CS, better mimic chondrocyte lacuna and create an appropriate environment for cartilage regeneration and functioning compared to the synthetic compounds [10]. CS is one of the main GAGs component found in a cartilage ECM that promotes cell differentiation and provides mechanical protection trough the tight cell-ECM interaction [11]. Moreover, it was shown that CS-tyramine hydrogels have a good cytocompatibility and the enhanced proliferation of porcine auricular chondrocytes [12]. Furthermore, CS can regulate cartilage metabolism and has anti-inflammatory properties trough cross-linking tyramine groups [13]. Gelatin (Gel), another natural compound, also possesses tyramine groups that could help to mimic the macromolecular structure and features of ECM [12]. Therefore, the enzyme-catalyzed reaction mediated by hydrogen peroxide and horseradish peroxidase (HRP) are non-cytotoxic, can easily crosslink both compounds and can control physical hydrogel properties by changing the concentration or polymerization time [14,15]. However, how CS, Gel and BM-MSCs respond to the mechanical load and affect OA cartilage explants was not investigated.

Physiologically articular cartilage is under persistent mechanical loads of various strengths, which is an important factor for cartilage homeostasis [16,17]. It is known that under physiological conditions, the compressive modulus of articular cartilage varies from 0.4–2.0 MPa [18], which is hardly achievable in vitro. Therefore, there are various types of mechanical compression systems in vitro mimicking the mechanotransduction system in vitro affecting cartilage during human walking or intensive sport [8,19]. However, the mechanical load mimicking systems in vitro strongly depend on various factors such as compression intensity and duration, used hydrogels and cells, size and origin of the cartilage explants and many others that cannot be equally evaluated. Therefore, if the mechanical load damages the functioning of the chosen experimental objects (hydrogels/cells, explants and other), it can be named as a mechanical overload, while a load positively affecting the cells or cartilage explants—as a mild one. The mild mechanical stimulation of chondrocytes was shown to increase the production of ECM [20], while the mechanical overload is damaging the cartilage tissue ECM and cells [17]. Therefore, the investigation of cartilage regeneration mechanisms applying tissue engineering tools such as injectable hydrogel and stem cells under the mechanical load both in vitro and in vivo is of importance.

In this study, we have incorporated BM-MSCs into CS-Tyr/Gel hydrogels and investigated their biocompatibility and chondrogenic differentiation under the not-damaging mechanical load. We have also investigated the effect of mechanical load on isolated human OA cartilage explants and their response to the co-culture with the CS-Tyr/Gel/BM-MSCs composite by evaluating the release of ECM components that are important to cartilage functioning, such as COMP and GAGs. The findings of this study suggest that the CS-Tyr/Gel/BM-MSCs composite can be used in investigating the regeneration mechanisms of OA cartilage in vitro with further translation in vivo.

## 2. Results

### 2.1. The Isolation and Characterization of Human BM-MSCs

The BM-MSCs were isolated from post-surgical biomaterial after obtaining the patients’ informed concern (Bioethics Committee Permission No. 158200-14-741-257). The BM-MSCs grew attached to the plastic and had a stromal cell-typical spindle form (Figure 1). The isolated BM-MSCs differentiated into MSC-typical adipogenic, osteogenic and chondrogenic directions were qualitatively identified by staining with oil red (red lipid drops), alizarin red (red calcium crystals) and collagen II (immunohistochemical staining), respectively (Figure 1A). The differentiation of the BM-MSCs to adipogenic and osteogenic directions was also quantitatively identified and compared to the not-differentiated cells (Figure 1B). Moreover, the BM-MSCs showed the MSC-typical surface biomarkers, analysed by a flow cytometer: were positive for the CD44, CD73, CD90 and CD105 (97 ± 2.5% of total cell population) and negative for the hematopoietic stem cell markers CD14, CD34 and CD45 (less than 5 ± 2.5% of total cell population).

### 2.2. Generation and Investigation of CS-Tyr/Gel/BM-MSCs Composites

The hydrogel and cell composite was prepared as written in the method. The polymerized CS-Tyr/Gel hydrogel with the BM-MSCs formed round-shaped transparent constructs that were easily seen under a light microscope (Figure 2A). The BM-MSCs were evenly spread in the polymerized CS-Tyr/Gel hydrogel (Figure 2B). In addition, the presence of the BM-MSCs and ECM in the CS-Tyr/Gel hydrogel after the incubation in chondrogenic medium for 21 days was identified histochemically by staining with toluidine blue-safranin O dyes (Figure 2C). Safranin O mainly stains highly sulphated glycosaminoglycans (chondroitin sulphate and keratan sulphate) and cell nuclei in red/orange, while toluidine blue stains proteoglycans and cells in blue. The total amount of BM-MSCs in CS-Tyr/Gel hydrogels was slightly higher at the 21st day of the incubation compared to the first one, revealing a good BM-MSCs survival in CS-Tyr/Gel and the eligibility of the CS-Tyr/Gel/BM-MSCs composite for cartilage regeneration studies (Figure 2D).

### 2.3. Chondrogenic Differentiation of CS-Tyr/Gel/BM-MSCs under the Mechanical Load

The investigation of chondrogenic differentiation of the CS-Tyr/Gel/BM-MSCs composite with or without the mechanical load could give additional information about the relevance of the composite for OA cartilage regeneration. The CS-Tyr/Gel/BM-MSCs composite was incubated in the chondrogenic differentiation medium for 14 days and then the mechanical load was applied for the next 7 days using the Flexcell FX-5000 system (Figure 3A) with FX-5000 plates (Figure 3B). The schematic presentation of the compression system is shown in Figure 3C. The mechanical load of 10 kPa, 1 Hz, 1 h/day for 7 days was chosen for the CS-Tyr/Gel/BM-MSCs composite in order to mimic the mild mechanical load on the human cartilage, which did not harm the BM-MSCs and stimulated the production of cartilage ECM. The chondrogenic differentiation of the BM-MSCs was initiated with TGF-β3, while the medium without TGF-β3 was used as a control (Figure 3D).

In addition, the chondrogenic differentiation of the BM-MSCs in CS-Tyr/Gel hydrogel with or without the mechanical load was identified by the immunohistochemical staining of ECM component collagen II (Figure 3G). Data show that CS-Tyr/Gel hydrogel provided not only a suitable environment for the 21 days of the BM-MSC survival in chondrogenic medium (Figure 3F), but even upregulated the level of collagen II under the mild mechanical load (Figure 3G, brown staining). The positive control for the immunohistochemical staining of collagen II (brown) is shown in the human cartilage explant (Figure 3E).

### 2.4. The Effect of Mechanical Load on OA Cartilage Explants

To investigate a negative effect of the mechanical load on OA cartilage explants, a stronger compression (30 kPa, 1 Hz, 1 h/day for 7 days) was applied to the human OA cartilage explants during the 7 days of incubation in chondrogenic medium (Figure 4).

The human OA cartilage was removed from the joint bone, cut into round circles as described in the method (Figure 4A), subjected to the mechanical load and levels of sulphated glycosaminoglycans and the proteoglycans were histochemically investigated using toluidine blue-safranin O staining in the cartilage explants: safranin O mainly stains highly sulphated glycosaminoglycans (chondroitin and keratan sulphate) red/orange, while toluidine blue stains proteoglycans in blue (Figure 4B). Data show that strong mechanical load, suppressed the total levels of ECM components glycosaminoglycans and proteoglycans compared to the not-compressed OA cartilage (Figure 4B).

In addition, the release of ECM components, such as COMP and GAGs, from the OA cartilage explants during the same mechanical load was investigated (Figure 5). The release of ECM components usually weakens the cartilage structure and makes it more susceptible to mechanical and other damaging and degenerating affects.

Data confirmed that a strong mechanical load not only decreased the total levels of GAGs and proteoglycans in OA cartilage (Figure 4), but also stimulated the release of cartilage important components, such as COMP (Figure 5A) and GAGs (Figure 5B). Surprisingly the chondrocytes survived the strong mechanical load in the OA cartilage explants (the lower LDH release shows better cell survival) (Figure 5C). Data suggest that ECM components in OA cartilage explants are more sensitive to the mechanical load than the chondrocytes and could serve as OA markers.

### 2.5. The Co-Incubation of CS-Tyr/Gel/BM-MSCs and OA Cartilage Explants

Since the injectable CS-Tyr/Gel/BM-MSCs composite survived the mechanical load and could be suitable for cartilage regeneration, it was interesting to investigate if the CS-Tyr/Gel/BM-MSCs composite could increase the total level of ECM components (Figure 6) and decrease the release of COMP and GAGs (Figure 7).

Therefore, the CS-Tyr/Gel/BM-MSCs composite was added on the top of the human OA cartilage explants and incubated in chondrogenic differentiation conditions for 21 days (Figure 6). The histochemical hematoxylin/eosin staining (cell nuclei are purplish-blue, and the extracellular matrix and cytoplasm are pink) shows that the BM-MSCs properly attached to the surface of OA cartilage explants (dash lines indicated an approximate boundary between the CS-Tyr/Gel/BM-MSCs and cartilage explants). The production of proteoglycans in CS-Tyr/Gel/BM-MSCs and cartilage explants was identified histochemically using toluidine blue staining (proteoglycans), while the level of collagen type II was identified immunohistochemically (brown) (Figure 6). The stainings of proteoglycans and collagen II were also confirmed as a good attachment or even some biointegration of the CS-Tyr/Gel/BM-MSC composite into the OA cartilage explants, i.e., the boundaries between the added CS-Tyr/Gel/BM-MSCs and the OA cartilage explants almost disappeared, particularly during the chondrogenic differentiation with TGF-β3 for 21 days (Figure 6, chondro medium + TGFβ3).

To prove the protecting effect of CS-Tyr/Gel/BM-MSCs on OA cartilage, the release of the ECM components, COMP and GAGs, was also measured after the incubation of OA cartilage explants with CS-Tyr/Gel/BM-MSCs in a chondrogenic medium with or without TGF-β3 for 21 days (Figure 7).

Data show that the release of COMP from the cartilage explants incubated in chondrogenic medium for 21 days without CS-Tyr/Gel/BM-MSCs was significantly higher compared to the cartilage explants incubated with CS-Tyr/Gel/BM-MSCs (Figure 7A). Data suggest that the CS-Tyr/Gel/BM-MSCs composite, added on the top of OA cartilage explants, can strengthen the OA cartilage by decreasing the release of the very important ECM components, COMP, particularly after 21 days of incubation (Figure 7A).

A similar situation was observed with the release of GAGs from the cartilage explants incubated with or without CS-Tyr/Gel/BM-MSCs in chondrogenic medium for 21 days (Figure 7B). Data show that the release of GAGs was significantly lower in the cartilage explants incubated with CS-Tyr/Gel/BM-MSCs compared to the explants without CS-Tyr/Gel/BM-MSCs incubation in chondrogenic medium for 21 days.

Data suggest that the injectable CS-Tyr/Gel/BM-MSCs composite positively affects OA cartilage by increasing cartilage proteoglycans and collagen II and decreasing the release of vitally important ECM components COMP and GAGs.

## 3. Discussion

Articular cartilage is a connective tissue that ensures joint movement and acts as a cushion to external mechanical stimuli [21]. Cartilage is an avascular tissue composed of chondrocytes surrounded by abundant ECM components [22]. Aging, trauma and excessive mechanical load are usually accompanied by cartilaginous tissue degradation and diseases, such as OA. Due to its avascular nature and the small number of chondrocytes, articular cartilage has a low ability to regenerate itself [23]. The treatment of articular cartilage lesions is challenging, due to complicated early diagnosis and drug delivery in quite later stages [1]. In addition, drug delivery is complicated because articular cartilage is an anisotropic tissue, presenting three parts with different collagen fiber arrangements [24]. The superficial zone with tightly packed and parallel aligned collagen fibers protects cartilage from various mechanical and other damages, the integrity of which is vitally important for proper joint functioning. Therefore, it is important to investigate new ways of protecting the surface of OA cartilage, as well as other layers.

ECM components in cartilage are synthesized by chondrocytes and maintain their proper functioning [25]. Cartilage ECM is mainly composed of GAGs, collagen II fibers and proteoglycans [8,25]. The composition and organization of ECM not only provides a specialized environment for the chondrocytes [26], but also determines the biomechanical properties of cartilage [27]. During the cartilage degeneration and OA progression, the ECM are gradually degraded, and components may serve as biomarkers of cartilage degeneration [28,29]. Cartilage COMP, also known as thrombospondin 5 (TSP5), is an ECM glycoprotein, which is associated with the breakdown of cartilage and is often used as a biomarker of OA [30,31]. COMP interacts with other ECM components and increases the mechanical strength of cartilage tissue [31]. It was shown that the release of COMP is associated with pathologic conditions, while its increased expression in cartilage is correlated with the increased levels of aggrecan and collagen type II and better cartilage integrity [32]. GAGs, another ECM component, are also crucial for the physiological functioning and mechanical resistance of cartilage. Chondroitin sulfate chains were shown to be the most abundant GAGs in adult articular cartilage, while keratan sulfate and HA only represent up to 30% of all GAGs in the cartilage [33]. The loss of GAGs is also an indicator of cartilage degeneration and deterioration as a response to the mechanical and other pathophysiological stimuli [34]. In this study, we have investigated the change of intracellular and extracellular ECM components in OA cartilage explants with or without the mechanical load and the effect of injectable hydrogel CS-Tyr/Gel and the BM-MSCs composite as a new type of cartilage regenerating tool protecting ECM components.

Hydrogels of different origin and composition are a promising approach for cartilage tissue repair [35]. Hydrogels are the class of elastic, three-dimensional, hydrophilic biomaterials able to absorb large amount of water while retaining their structure and displaying the smooth surface necessary for the proper joint functioning [36]. Hydrogels for the repair of articular cartilage should also be biocompatible, biodegradable and be able to withstand mechanical compression in vivo [37]. Various types of natural and synthetic hydrogels and/or their combinations and ways of polymerization were used for the purpose of regenerating cartilage tissue [38]. However, the most commonly used and promising are natural polymers, such as collagen, chondroitin sulphate, fibrin, silk fibroin and others, with or without synthetic components [38,39]. In addition, among the various types of hydrogels, injectable hydrogels have also demonstrated great potential, as three-dimensional systems in cartilage tissue engineering, due to their framework mimicking cartilage lacuna, possessing a good penetration of nutrition and cell survival, having minimal invasive properties during injection into the articular joint, and having the ability to match irregular cartilage defects [40]. Besides enzymatic polymerization, the next generation of biomatrices based on layer-by-layer (LbL) ultrathin films [41], bioinks with nanoparticles [42] or magnetically-assisted 3D bioprinting constructs [43] are also of interest as future potential cartilage regeneration strategies.

It has been reported that injectable the CS-based hydrogel system promoted the chondrogenesis of rat adipose-derived MSCs and upregulated the expression of Sox9, aggrecan (ACAN) and collagen II (COL2A1) [44]. The CS/hyperbranched poly(ethene glycol) (PEG)-based hybrid injectable hydrogel system (HB-PEG), made by the “click” thiol–ene reaction, had good biocompatibility and mechanical properties. BM-MSCs, cultured in CS and poly (γ-glutamic acid) (CS-γ-PGA) hydrogels in vitro, also secreted more chondrocyte exosomes as compared with BM-MSCs cultured in a Petri dish. However, CS-γ-PGA hydrogels had a rapid degradation (25–60 h depending on the used concentration) [45]. Additional research demonstrated the potency of CMC-MC-P (carboxymethyl chitosan methylcellulose pluronic) and zinc chloride hydrogels containing meloxicam and nanoparticles. The CMC-MC-P hydrogel with nanoparticles increased the metabolic activity and attachment of sheep chondrocytes in vitro [46]. It was also shown that the injectable hydrogel system with PLGA (poly (lactic-co-glycolic acid) microspheres and BMP-2 (bone morphogenetic protein-2) improved BM-MSCs-induced cartilage lesion repair in rabbits [47]. Another study showed that hydrogels composed of injectable chitosan, Pluronic-F127, crosslinking agent tripolyphosphate and dexamethasone remained for a long time in the joint injection site in mice [48]. However, many studies in vitro and in vivo used injectable hydrogels with various additional bioactive compounds but without a biomechanical test.

The hydrogels and cells are not the only components important for cartilage regeneration. It is agreed that the mechanical load plays an important role in cartilage homeostasis and chondrocyte mechanotransduction processes and is also a key factor determining the development of OA [8,49]. In this study, we have investigated the biocompatibility and chondrogenic differentiation of BM-MSCs encapsulated into the main GAGs component chondroitin sulphate-based hydrogel under a mild (not damaging cells) mechanical load (10 kPa (2 s)/5 kPa (1 s), 1 Hz, 1 h/day) for 7 days. The mechanical load increased the level of collagen type II in CS-Tyr/Gel/BM-MSCs composite, suggesting its suitability for cartilage regeneration. Similar studies also reported that dynamic compression by 10% strain, 1 Hz, 1 h/day improved the chondrogenesis of human chondrocytes and BM-MSCs cultured in silk fibroin and silk fibroin with gelatin/chondroitin sulfate/hyaluronate scaffolds and increased the production of collagen II and expression of other ECM components [50]. In addition, the data of this study showed that the mechanical load negatively affected human OA articular cartilage explants by releasing ECM proteins, such as GAGs and COMP, but did not affect the viability of the chondrocytes. The findings suggest that the ECM components are more sensitive to the mechanical load compared to the chondrocytes located in lacuna.

Therefore, it is important to combine the most important factors for the cartilage regeneration in vitro and in vivo: the best suited stem cells and hydrogels, their biocompatibility, cartilage regenerating properties and mechanical load positively affecting the OA cartilage. However, there are many different mechanical load systems in vitro, that strongly depend on the used equipment, hydrogels, cells, duration of the load and other experimental conditions that can hardly be compared with each other. For example, it was shown that BM-MSCs encapsulated into the collagen hydrogels and mechanically stimulated under static or dynamic conditions (a 10% peak compressive sinusoidal strain at 1Hz frequency, for 2 h/day for 21 day) could activate the chondrogenic differentiation-related markers, collagen II, aggrecan and Sox9 [51], while another study showed that a similar mechanical load on synovial MSCs encapsulated into agarose hydrogel under the similar conditions downregulated the expression level of chondrocyte-specific markers [52]. Therefore, the proper chose of experimental mechanical load conditions is not less important than used hydrogel/cell composites for the OA cartilage regeneration studies that can differently influence the ECM integrity and chondrocyte homeostasis affecting the articular cartilage tissue studies both in vitro and in vivo.

## 4. Materials and Methods

### 4.1. Preparation of Cartilage Explants and Culture under Mechanical Load

Fragments of human OA articular cartilage were obtained from patients undergoing joint replacement surgery at Vilnius Santaros Hospital (bioethics committee permission No 158200-14-741) in a sterile container with PBS (Sigma Aldrich, St. Louis, MO, USA). All further work was carried out under sterile conditions. The cartilage was washed several times with PBS containing 2% penicillin/streptomycin (PS) (Thermo Fisher Scientific, Waltham, MA, USA) and transferred to a Petri dish. The cartilage was cut into flat, round explants of 3 mm diameter and 3 mm height and each explant was weighted. The cartilage explants were washed and transferred into two wells of six well plates (120 mg of cartilage into one well), i.e., one explant for the mechanical load by the Flexcell FX-5000 system, while another one was used as a control explant. Cartilage explants were cultured in chondrogenic medium, which was composed of high glucose (4.5 g/L) DMEM medium, 1% PS, 1% insulin-transferrin-selenium (Thermo Fisher Scientific, Waltham, MA, USA), 350 μM L-proline (Carl Roth GmbH, Karlsruhe, Germany), 0.1% dexamethasone and 170 μM ascorbic acid-phosphate (Sigma Aldrich, St. Louis, MO, USA).

The next day, the plate with cartilage explants was transferred to the mechanical compression system, Flexcell FX-5000, situated in the incubator with 5% CO_2_ and 37 °C and compressed using a square-shaped dynamic compression mode with 30 kPa, 1 Hz frequency strength, 1 h/day for 7 days. The chondrogenic medium was changed every second day. After 7 days, the explants from each study group, with or without the mechanical load, were taken for further histological, immunohistochemical, COMP, GAG and LDH assays.

Chondrocyte viability in explants was analyzed by the released amount of lactate dehydrogenase (LDH) (Thermo Fisher Scientific, Waltham, MA, USA) according to the manufacturer’s recommendations, using spectrophotometer SpectraMax i3 (Molecular Devices, CA, USA). LDH is directly related to the damage of cells’ plasma membranes, i.e., the lower LDH release represents a better cell viability.

### 4.2. Isolation and Characterization of BM-MSCs

The BM-MSCs were isolated from healthy human bone marrows (n = 5). The human bone marrow samples remaining after the articular surgery were received from the Vilnius University Hospital Santaros Clinics, according to bioethics committee permission No 158200-14-741. The type and size of the obtained bone sample depended on the type of bone injury and surgery. Bone samples were washed with PBS, disinfected with ethanol, transferred to a sterile dish and once more washed with PBS containing 2% PS. The bone marrow was extracted from the bone and placed in a Petri dish with DMEM medium (1 g/L glucose). Then, the bone marrow was chopped to a liquid consistency and mixed in DMEM medium. The obtained suspension was filtered through 100 µm filter and centrifuged for 10 min at 350 g. The cell pellet was resuspended in complete DMEM medium, supplemented with 10% fetal bovine serum (FBS) (Merck Millipore, Merck KGaA, Darmstadt, Germany), 1% PS and fibroblast grow factor 2 (FGF-2) (20 ng/mL) (Thermo Fisher Scientific, Waltham, MA, USA), counted and cultured in flasks under regular cell growth conditions (5% of CO_2_ and 37 °C). Cells were harvested with trypsin/EDTA (Thermo Fisher Scientific, Waltham, MA, USA) solution and the complete DMEM medium was changed twice a week.

The identification of MSC-typical surface biomarkers was performed with a flow cytometer. Briefly, the BM-MSCs were harvested using trypsin, centrifuged, resuspended in PBS supplemented with 2% BSA (Sigma Aldrich, St. Louis, MO, USA) and incubated on ice for 30 min. After that, the cells were washed with PBS and incubated with antibodies against: type 1 transmembrane glycoprotein (CD 44-IgG2b-FITC, 555478, BD Biosciences, Waltham, San Jose, CA, USA) endoglin (CD105-ImmunoglobulinG1-Allophycocyanin (MHCD10505, Thermo Fisher Scientific, Waltham, MA, USA); thymocyte differentiation antigen 1 (CD90-ImmunoglobulinG1-Fluorescein isothiocyanate (328108, BioLegend, San Diego, CA, USA)); ecto-5′-nucleotidase (CD73-ImmunoglobulinG1-Fluorescein isothiocyanate (561254, BD Biosciences, San Jose, CA, USA)); protein tyrosine phosphatase, receptor type, C (CD45-ImmunoglobulinG2a-Fluorescein isothiocyanate (sc-70686, Santa Cruz Biotechnology, Dallas, TX, USA)); stage-specific embryonic antigen 1 (CD15-Immunoglobu linG1-Phycoerythrin (555402, BD Biosciences, San Jose, CA, USA)); and transmembrane phosphoglycoprotein protein (CD34-ImmunoglobulinG1-Fluorescein isothiocyanate (1F-297-T100, Exbio, Praha, Czech Republic)). After incubation, the cells were washed, centrifuged, resuspended in PBS supplemented with 2% BSA and analyzed with a flow cytometer.

The BM-MSCs differentiated into osteogenic and adipogenic lineages were induced using specific cocktails for each differentiation. BM-MSCs were seeded into a 12 well plate (60,000 cells/well) for adipogenic and (40,000 cells/well) for osteogenic differentiation. Both differentiations were induced when cells reached sub-confluence. Adipogenic medium consisted of low glucose (1 g/L) DMEM medium, 1% PS, 20% FBS, 1 µmol/L dexamethasone (Sigma Aldrich, St. Louis, MO, USA), 60 µmol/L indomethacin (Sigma Aldrich, St. Louis, MO, USA) and 50 µmol/L 3-isobutyl-1-methylxanthine (IBMX) (BioSource, San Diego, CA, USA) and cells were incubated for 21 days. After differentiation, lipid droplets formed in cells were stained with oil red-o dye (Carl Roth GmbH, Karlsruhe, Germany) and visualized under the light microscope. After visualization, lipid droplets were dissolved in 70% isopropanol solution (Eurochemicals, Vilnius, Lithuania), collected into a 96 well plate and their absorbance measured using a spectrophotometer (absorbance, 520 nm).

The osteogenic differentiation medium consisted of high glucose (4.5 g/L) DMEM medium (Thermo Fisher Scientific, Waltham, MA, USA), 1% penicillin/streptomycin, 10% FBS, 0.1 µmol/L dexamethasone, 50 µg/mL ascorbic acid and 10 mmol/L β-glycerophosphate (Santa Cruz Biotechnology, Dallas, TX, USA). The osteogenic differentiation was performed for 21 days and the calcium deposition was evaluated by staining of BM-MSCs with alizarin red S (Carl Roth GmbH, Karlsruhe, Germany). After that, the stained calcium crystals were dissolved in 10% cetylpyridinium chloride (Sigma Aldrich, St. Louis, MO, USA), collected into a 96 well plate and their absorbance measured using spectrophotometer (absorbance, 562 nm).

Chondrogenic differentiation of BM-MSCs was induced by using a chondrogenic medium, composed of high glucose (4.5 g/L) DMEM medium, 1% PS, 1% insulin-transferrin-selenium (Thermo Fisher Scientific, Waltham, MA, USA), 350 μM L-proline (Carl Roth GmbH, Karlsruhe, Germany), 0.1% dexamethasone, 170 μM ascorbic acid-phosphate (Sigma Aldrich, St. Louis, MO, USA) and 10 ng/mL TGF-β3 (Thermo Fisher Scientific, Waltham, MA, USA). The 250,000 cells/15 mL tube was used for the 3D chondrogenic differentiation and evaluated by the immunohistochemical staining of collagen II. BM-MSCs were differentiated for 21 days. Three technical replicates were used for each donor and undifferentiated cells were used as controls for immunostaining.

### 4.3. Synthesis of CS-Tyr

Tyramine (Tyr) was conjugated to chondroitin-4-sulfate sodium (CS) through EDC/NHS (N-ethyl-N′-(3-(dimethylamino)propyl)carbodiimide/N-hydroxysuccinimide) reaction (Figure 8).

Briefly, 500 mg of chondroitin-4-sulfate was dissolved in 10 mL of MES buffer solution (pH 4.5, 100 mM), 141.9 mg of tyramine was dissolved in 20 mL of MES buffer solution (pH 4.5, 100 mM, Sigma Aldrich, St. Louis, MO, USA), 197.7 mg 1-Ethyl-3-(3-dimethylaminopropyl) carbodiimide (EDC) and 119.2 mg N-hydroxysuccinimide (NHS) was dissolved in 10 mL of MES buffer solution (pH 4.5, 100 mM) and then added to the chondroitin-4-sulfate solution. After 30 min, all solutions were mixed and the pH was adjusted to 4.5. The reaction mixture was stirred in the dark at room temperature (RT, 24 °C) for 24 h. The resulting solution was purified in a dialysis membrane (MWCO 3500 Da) against acidified deionized water (pH ~3) four times under gentle stirring and finally against deionized water (pH ~7) for 2 h under vigorous stirring. The final product was lyophilized and stored at −20 °C.

For the chondroitin sulfate-tyramine hydrogel preparation, the chondroitin sulfate-tyramine was dissolved in PBS at RT. The pH was adjusted to 7.4 by adding sodium hydroxide solution. The HRP solution was prepared by diluting the HRP stock solution in PBS to 10 U/mL. The H_2_O_2_ solution was prepared by dissolving H_2_O_2_ in PBS to make 100 mM H_2_O_2_ solution. CS-Tyr hydrogel was prepared by mixing the chondroitin sulfate-tyramine solution, HRP solution and H_2_O_2_ solution.

### 4.4. Formation and Investigation of CS-Tyr/Gel Hydrogel

The formation of the CS-Tyr/Gel hydrogel was mediated by an enzyme-induced oxidative coupling reaction as written in [14], with some modifications, which connects the phenolic functionality between CS-Tyr and gelatin. Briefly, 8% of chondroitin sulfate-tyramine and 4% of gelatin (Extra Pure, SLR (Thermo Fisher Scientific, Waltham, MA, USA) were separately dissolved in 10× PBS at RT, and CS-Tyr pH was adjusted to 7.4. Then, two mixtures were prepared: (1) the 8% of CS-Thyr and 0.6 U/mL of peroxidase from horseradish (HRP) (Sigma Aldrich); (2) the 4% of gelatin and 6 mM of hydrogen peroxide solution ≥ 30% (H_2_O_2_) (Sigma Aldrich, St. Louis, MO, USA). The prepared two mixtures were mixed in a 1:1 ratio and allowed to set at RT for 10 min. The gelation time of the mixed hydrogel components was measured by the inversed vial test—the properly set hydrogel remained in the vial.

Hydrogel consisting of CS-Tyr and a varying amount of gelatin and their elasticity were analyzed using dynamic mechanical analysis (modular compact rheometer, MCR-102, Anton Paar, Graz, Austria). The mechanical test was operated in frequency sweep mode of 0.1–10 Hz at 1% strain and 37 °C. The elasticity of the CS-Tyr/gelatin hydrogels was not affected by an addition of a small amount of gelatin (Figure 9). When the gelatin content increased to 1%, the elastic modulus decreased by approximately 50%. However, the elastic moduli did not further decrease as the gelatin content increased to 4%. The decrease in elastic moduli might be due to a low tyrosine content of gelatin. Since gelatin supports cell activity, 4% gelatin was used in the subsequent cell experiments. The swelling ratio of the CS-Tyr/gelatin hydrogel, based on its dried weight, was approximately 30~35 folds.

For the incorporation of BM-MSCs into CS-TYR/Gel hydrogel, 250,000 of harvested cells were washed with PBS, centrifuged and suspended into the first hydrogel mixture. Then, two mixtures were mixed in a 1:1 ratio and polymerized in the six well Flexcell plate at RT for 10 min. The regular growth medium was added to the polymerized CS-Tyr/Gel composites, plated and transferred to a 37 °C and 5% CO_2_ incubator for 14 days. The medium was changed twice a week. Then, the hydrogels were subjected to a mechanical load (10 kPa, 1 Hz, 1 h/day for the next 7 days) and further investigated with various assays.

### 4.5. Chondrogenic Differentiation of BM-MSCs Encapsulated in CS-Tyr/Gel

The BM-MSCs were encapsulated into CS-Tyr/Gel hydrogels and transferred into two different six well plates (50 × 10^3^ of the cells in 50 μL of CS-Tyr/gel for one well). One plate was used for the mechanical load with a commercial Flexcell FX-5000 system (Flexcell^®^ International Corporation, Burlington, NC, USA), while the other was used as a control.

Chondrogenic medium composed of high glucose (4.5 g/L) DMEM medium, 1% PS, 1% insulin-transferrin-selenium (Thermo Fisher Scientific, Waltham, MA, USA), 350 μM L-proline (Carl Roth), 0.1% dexamethasone, 170 μM ascorbic acid-phosphate (Sigma Aldrich, St. Louis, MO, USA) and 10 ng/mL TGF-β3 (Thermo Fisher Scientific, Waltham, MA, USA) was added on the same day (3 mL/well). The plates with hydrogels/cells were divided into two groups (three wells/group): (1) control (hydrogels/cells with a chondrogenic medium without TGF-β3); (2) differentiated (hydrogels/cells with a chondrogenic medium and with growth factor TGF-β3). Two weeks after the chondrogenic differentiation, the Flexcell plate with hydrogels/cells was transferred into the Flexcell FX-5000 system and compressed using a square shaped dynamic function with 10 kPa, 1 Hz compression, 1 h/day for 7 days. The control hydrogels were not mechanically compressed. The chondrogenic medium was changed three times a week. After 7 days of stimulation, the Flexcell and control plates with hydrogels/cells were analyzed immunohistochemically according to collagen type II labeling (see Section 4.9).

The biocompatibility of CS-Tyr/Gel and BM-MSCs was investigated at first and the last day of chondrogenic differentiation by using the metabolic CCK-8 assay according to the manufacturer’s description.

### 4.6. The Effect of CS-Tyr/Gel/BM-MSCs on Human OA Cartilage Explants

After the isolation of the cartilage explants and overnight incubation in growth medium, the explants were transferred into 12 well plates (120 mg/well) and divided into three groups: (1) four wells were incubated with DMEM medium; (2) four wells were incubated with chondrogenic medium; (3) four wells were incubated with chondrogenic medium + TGF-β3 (10 ng/mL). Two wells of each group of cartilage explants were exposed to the CS-Tyr/Gel/BM-MSC composite (250 k of BM-MSCs in 250 μL of CS-Tyr/Gel) and incubated for 21 days. The media were collected after 3, 9 and 21 days for a COMP release analysis, and GAG release was analyzed only after 21 days. After 21 days, the cartilage explants with the CS-Tyr/Gel/BM-MSC composite from each study group were histologically (hematoxylin and eosin, toluidine blue) and immunohistochemically (collage type II labeling) evaluated.

### 4.7. Secretion of Cartilage Oligomeric Matrix Protein (COMP)

The secretion of COMP was measured in the supernatants of: (1) cartilage explants incubated with and without CS-Tyr/Gel/BM-MSCs composites for 3, 9 and 21 days; (2) cartilage explants after the mechanical load for 7 days. The supernatants were collected from all samples 3 days after the medium change and the levels of COMP were detected using COMP ELISA (BioVendor, Karasek, Brno Czech Republic) according to the manufacturer’s instructions. The supernatants of CS-Tyr/Gel/BM-MSCs were not diluted, while the supernatants of the cartilage explants were diluted by 500-fold. The absorbance was measured at 450 nm using the spectrophotometer, SpectraMax i3.

### 4.8. Release of Glycosaminoglycans (GAGs)

The release of glycosaminoglycans (GAGs) was assessed in the supernatants of: (1) cartilage explants incubated with and without the CS-Tyr/Gel/BM-MSC composites for 21 days; (2) cartilage explants after the mechanical load for 7 days. Measurements were performed using colorimetric 1,9-dimethylmethylene blue dye (Bicolor Life Science assays, Carrickfergus, UK) dye, according to the manufacturer’s instructions. The absorbance was measured at 656 nm using the spectrophotometer, SpectraMax i3.

### 4.9. Histochemical and Immunohistochemical Analysis of Cartilage Explants and CS-Tyr/Gel/BM-MSCs Composites

For the histochemical analysis of the CS-Tyr/Gel/BM-MSC composites, the samples were fixed in 10% of neutral formalin and embedded into paraffin. The 4-micrometer sections were deparaffinized and stained with Toluidine Blue-Safranin-O (pH 2.0) for 3 min. The safranin O stains glycosaminoglycans (GAGs) and cell nuclei orange/red, while toluidine blue identifies cells by staining them blue.

The immunohistochemical staining of CS-Tyr/Gel/BM-MSCs and cartilage explants with antibodies against collagen type II (Abcam, Cambridge, UK) was performed after the antigen retrieval with a citrate buffer pH 6.0 at +98 °C for 20 min and endogenous peroxidase blocking with 0.3% hydrogen peroxide for 15 min at RT. The avidin–biotin complex (ABC) staining kit (Santa Cruz Biotechnology, Dallas, TX, USA) and 3.3-diaminobenzidine as a chromogen were used. Stained sections were evaluated and blindly scored independently by two histology experts.

### 4.10. Statistics

The results are presented as mean ± standard deviation (SD) from three repeats of no less than three cell cultures. Data are considered significant at *p* ≤ 0.05 calculated by MS Excel and Graphpad Prism 7 softwares.

## 5. Conclusions

In this study, we have investigated the relevance of an injectable composite of CS-Tyr/Gel hydrogel with encapsulated BM-MSCs for OA cartilage regeneration in vitro by direct its application on OA cartilage explants. Hereby we demonstrated excellent adhesion and/or biointegration of the hydrogel/cell composite on OA cartilage explants. Data showed that the ECM components of OA cartilage explants are more susceptible to a mechanical load than chondrocytes. However, the CS-Tyr/Gel/BM-MSC composite, placed on the top of human OA cartilage explants and incubated in chondrogenic medium for 21 days, significantly decreased the release of the important cartilage ECM components, COMP and GAGs. The strongest OA cartilage protecting effect by the CS-Tyr/Gel/BM-MSC composite was observed at the 21st day of its application, which suggests that it has long-term beneficial effects on cartilage ECM. To achieve a better resistance to mechanical load, an even longer incubation period may be required to strengthening cell–cartilage adhesion and/or biointegration of the hydrogel/cell composite on OA cartilage explants.

In summary, the data showed that the ECM of human OA cartilage explants is susceptible to the mechanical load, and the CS-Tyr/Gel/BM-MSC composite can be successfully applied for the OA cartilage protection and regeneration studies in vitro, as well as seeming a promising candidate for the further translation to novel therapies of OA in vivo.

## 6. Perspectives

CS-tyramine-gelatin hydrogels are injectable type of hydrogels and can be successfully used to deliver not only BM-MSCs but also other types of the cells or biocompounds into OA cartilage for the investigation of both cartilage regeneration and protection mechanisms in vitro and further therapeutic translation in vivo.

However, the efficiency of the CS-Tyr/Gel/BM-MSC composite application for the OA cartilage regeneration still needs additional experiments in vitro, particularly during the long-term incubations with/without the mechanical load or additional chondrogenic differentiation stimulating biomolecules. Further studies that improve the composition of CS-Tyr/Gel hydrogel and/or selecting the best regenerative potential cells for the OA cartilage regeneration are needed, as well as a broader evaluation of the secreted and intracellular cartilage compounds as potential biomarkers of OA development.

## Figures and Tables

**Figure 1 ijms-24-03451-f001:**
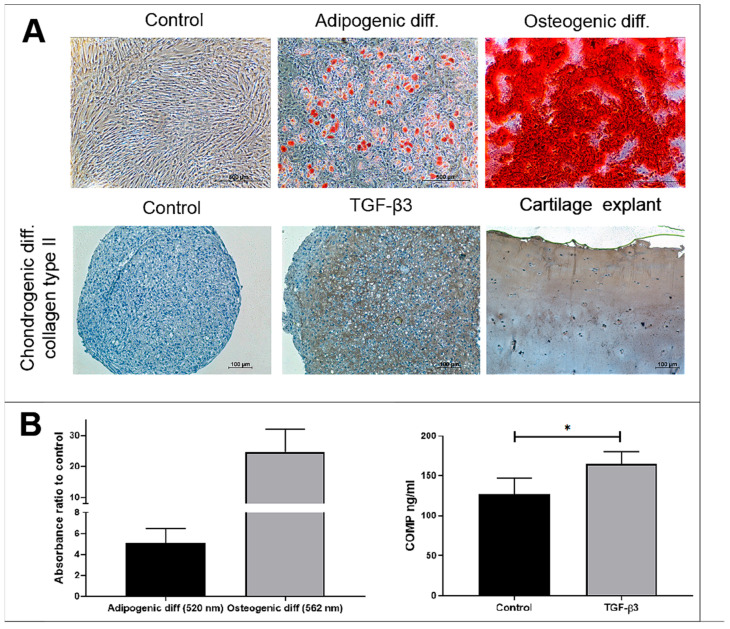
Adipogenic, osteogenic (2D) and chondrogenic (3D) differentiation of BM-MSCs. (**A**) The qualitative evaluation of the adipogenic, osteogenic and chondrogenic differentiation of BM-MSCs by incubating with the appropriate differentiation media for 21 days. The adipogenic differentiation was identified by staining the BM-MSCs with oil red (red lipid droplets), osteogenic—with alizarin red (red calcifications) and chondrogenic—by immunohistochemical labelling with anti-collagen II antibodies (brown) and visualized using a light microscope, by ×40 and ×100. Control chondrogenic pellets were incubated without TGF-β3. Cartilage tissue is provided as a positive control for collagen type II immunohistochemical labelling. (**B**) The quantitative evaluation of adipogenic and osteogenic differentiations of BM-MSCs (21 days), dissolving lipid droplets in isopropanol and calcium crystals in cetylpyridinium chloride, respectively. Absorbance was measured at 520 nm and 562 nm, respectively. The control cells for the adipogenic and osteogenic differentiation were incubated with a complete DMEM medium and stained with the same dyes. Data are presented as a ratio to control (not differentiated) cells and are shown as mean ± SD from no less than three repeats of three cell cultures. For the quantitative chondrogenic differentiation—the secretion of COMP protein by BM-MSCs pellets with and without TGF-β3 is provided. Data are presented as mean ± SD and are significant at * *p* ≤ 0.05 from no less than three repeats of three pellets.

**Figure 2 ijms-24-03451-f002:**
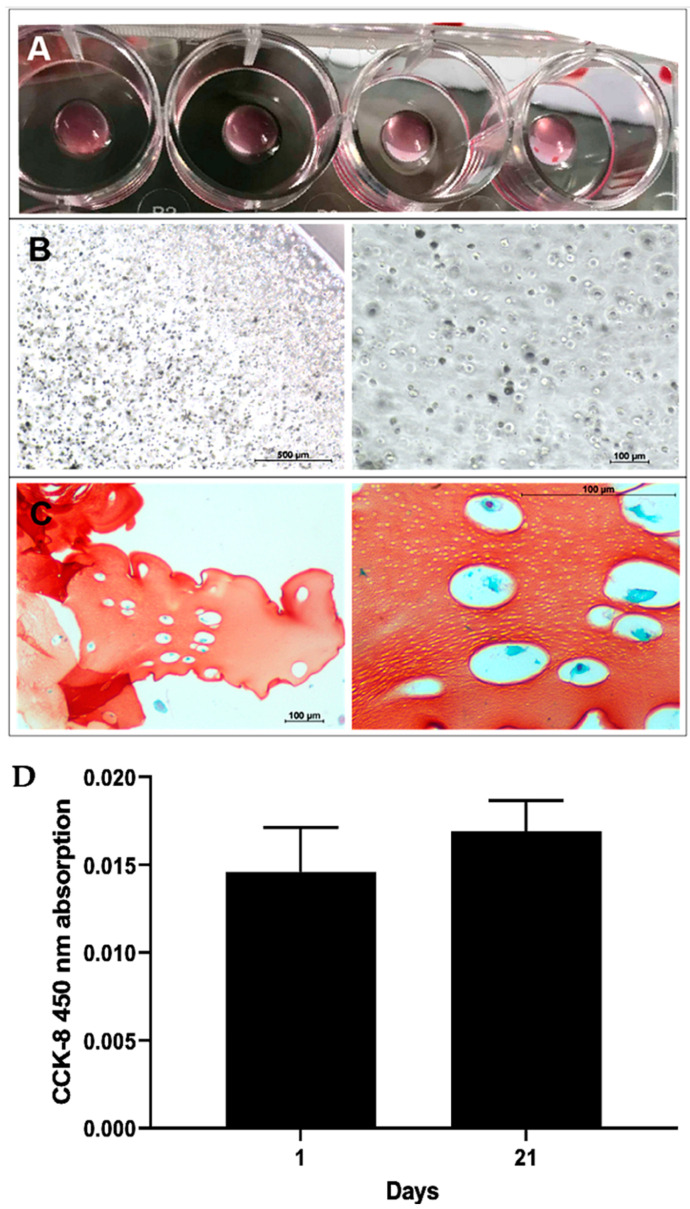
The generation and investigation of the CS-Tyr/Gel/BM-MSCs composite. (**A**) The micrograph of the polymerized CS-Tyr/Gel/BM-MSCs composite. (**B**) The light microscope micrographs of BM-MSCs encapsulated into CS/Tyr/Gel hydrogel. (**C**) Toluidine blue-safranin O staining of paraffin sections of the CS-Tyr/Gel/BM-MSCs composite after the incubation in chondrogenic medium for 21 days. (**D**) The viability of BM-MSCs after the encapsulation into CS-Tyr/Gel and incubation in chondrogenic medium for 1 and 21 days, measured with CCK-8 kit. Data are presented as mean ± SD from not less than three repeats of three cell cultures. The representative micrographs of the hydrogel/cell composite and histological staining are shown.

**Figure 3 ijms-24-03451-f003:**
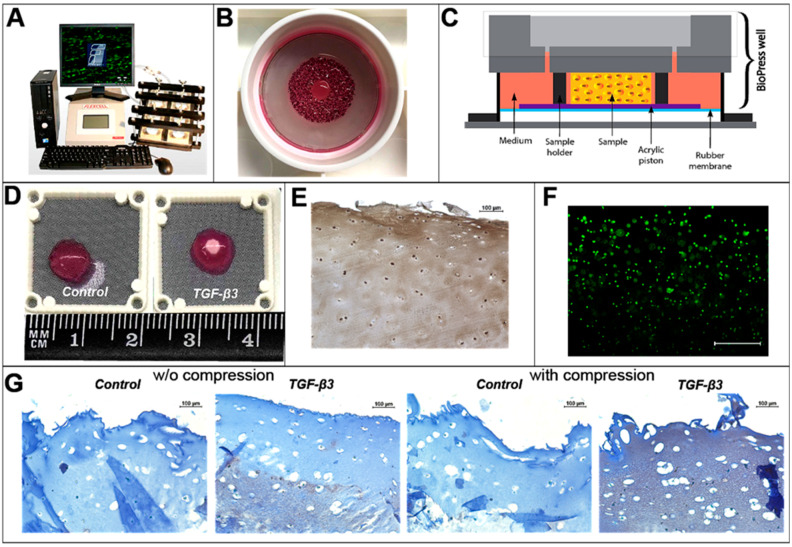
The chondrogenic differentiation of BM-MSCs incorporated into CS-Tyr/Gel hydrogels for 21 days of chondrogenic differentiation with and without the mechanical load. (**A**) Flexcell FX-5000 system. (**B**) Flexcell plate with the CS-Tyr/Gel/BM-MSCs. (**C**) Schematic representation of Flexcell compression. (**D**) The CS-Tyr/Gel/BM-MSCs with and without the TGF-β3 21 days after the chondrogenic differentiation. (**E**). Immunohistochemical positive staining control of collagen type II in human cartilage explant. (**F**) Viability of BM-MSCs in CS-Tyr gel at the 21st day of chondrogenic differentiation stained with live/dead kit and visualized with fluorescent microscope (×40). (**G**) Immunohistochemical collagen type II staining in CS-Tyr/Gel/BM-MSCs composite 21 days after the chondrogenic differentiation with or without the mechanical load mimicking a human walk. The CS-Tyr/Gel/BM-MSCs composite was first incubated in the chondrogenic medium for 14 days, then the mechanical load (10 kPa, 1 Hz, 1 h/day) was applied for the next 7 days of differentiation.

**Figure 4 ijms-24-03451-f004:**
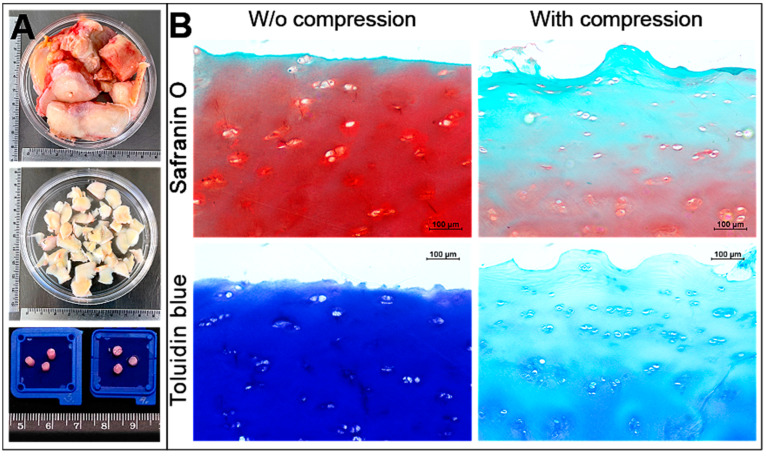
The effect of mechanical load on human OA cartilage explants. (**A**) The macroscopic view of received parts of human joint, chopped cartilage and prepared the explant circles for further experiments. (**B**) Histochemical staining of proteoglycans with safranin O (red/orange) and with toluidine blue (blue) with/without mechanical compression (30 kPa, 1 Hz, 1 h/day for 7 days).

**Figure 5 ijms-24-03451-f005:**
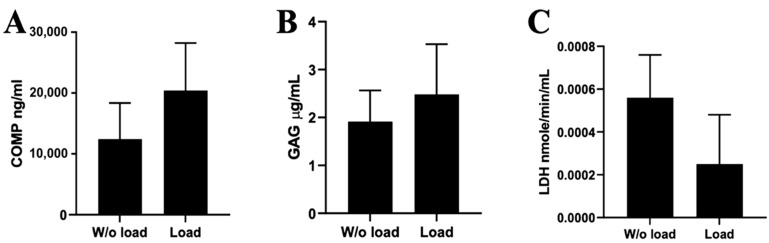
The release of ECM components during mechanical load of OA cartilage explants. (**A**) The release of cartilage oligomeric matrix protein (COMP); (**B**) The release of glycosaminoglycans (GAGs); (**C**) The release of lactate dehydrogenase activity (LDH) revealing cell death after the mechanical load for 7 days (30 kPa (2 s)/5 kPa (1 s), 1 Hz, 1 h/day). Data are presented as mean ± SD from no less than three repeats of three cell cultures.

**Figure 6 ijms-24-03451-f006:**
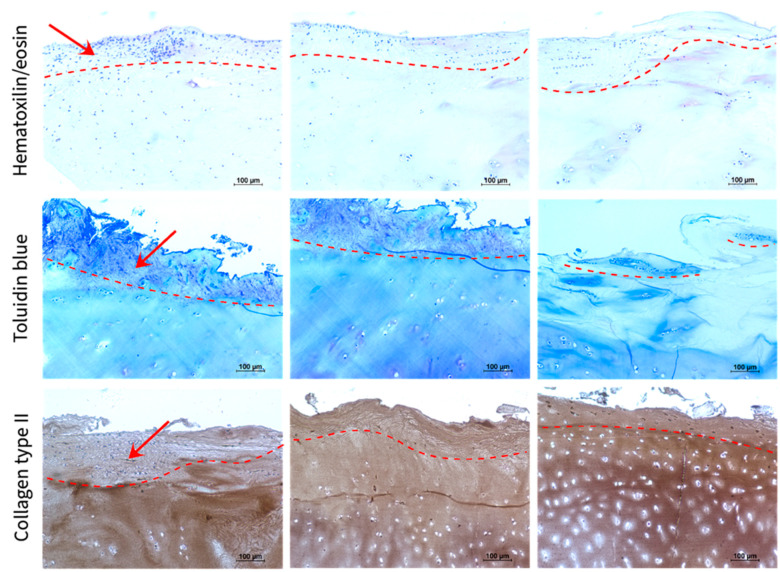
Histochemical and immunohistochemical analysis of OA cartilage explants coated with CS-Tyr/Gel/BM-MSC hydrogel, cultured in DMEM medium, chondrogenic medium and chondrogenic medium with TGF-β3 for 21 days. Cells in the CS-Tyr/Gel and cartilage tissue were identified by hematoxylin/eosin staining (cell nucleus are dark blue, other structures-pink). Toluidine blue identifies proteoglycans (blue), and collagen type II was identified immunohistochemically (brown). Red arrows show the added CS-Tyr/Gel/BM-MSCs, while the dash line shows an approximate boundary of added CS-Tyr/Gel/BM-MSCs composite after 21 days of incubation.

**Figure 7 ijms-24-03451-f007:**
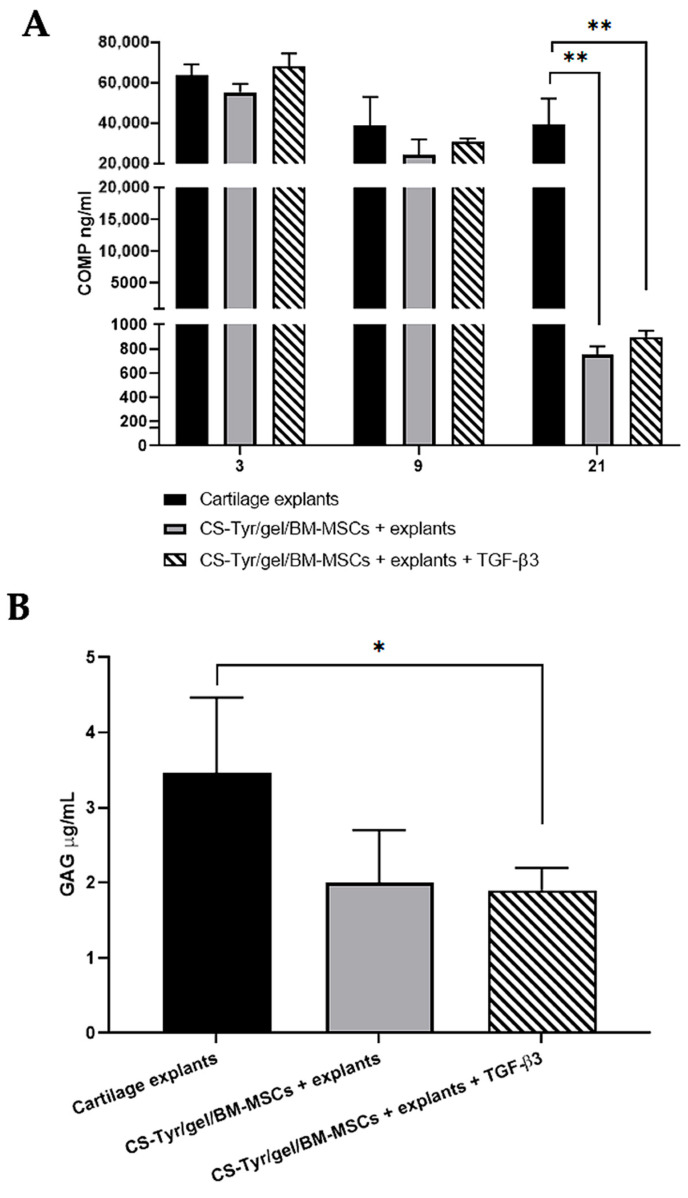
The release of ECM components from the human OA cartilage explants with and without the CS-Tyr/Gel/BM-MSCs during the 21 days of incubation chondrogenic medium. (**A**) The release of the cartilage oligomeric matrix protein (COMP) from the cartilage explants without and with the CS-Tyr/Gel/BM-MSCs hydrogel; incubated in chondrogenic medium with and without TGF-β3 for 3, 9 and 21 days. (**B**). The release of glycosaminoglycans (GAGs) from the human OA cartilage explants with or without the CS-Tyr/Gel/BM-MSC composite, incubated in chondrogenic medium with or without TGF-β3 for 21 days. Data are presented as mean ± SD and are significant at * *p* ≤ 0.05 and ** *p* ≤ 0.01 from no less than three repeats of three cell cultures.

**Figure 8 ijms-24-03451-f008:**
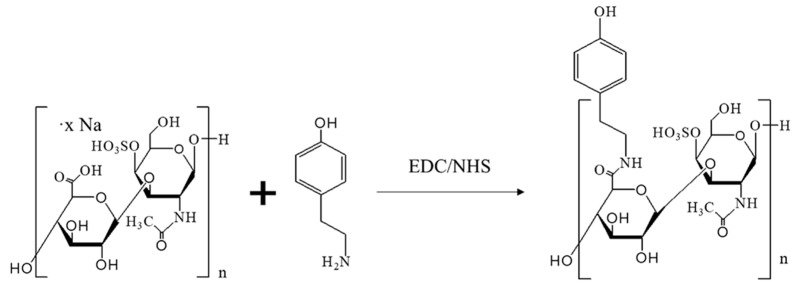
Chemical synthesis of CS-Tyr through EDC/NHS reaction.

**Figure 9 ijms-24-03451-f009:**
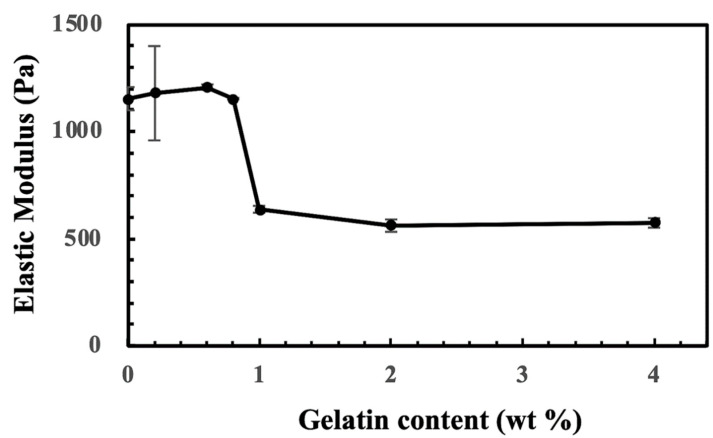
The elastic moduli of CS-Tyr/gelatin hydrogels, containing 4% CS-Tyr and 0–4% of gelatin. n = 3, error bars represent standard deviations.

## Data Availability

The data supporting these findings can be found at the State Research Institute Centre for Innovative Medicine, Department of Regenerative Medicine.

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
