# Peer review of "Chondroitin Sulfate-Tyramine-Based Hydrogels for Cartilage Tissue Repair"

_ijms, 2023, doi:10.3390/ijms24043451_

Round 1
Reviewer 1 Report
1. There are some formatting errors in the article. For example, Line 20 and Line 81, Gelatin not Gelatine.Spelling of references must be checked to meet the journal style (such as Reference 15). Please check carefully and use abbreviation properly.
2. Figure 8, gelatin itself also has certain tyramine residues. Therefore, is it a combination of complexed gelatin and complexed CS when the HRP enzyme is cross-linked?
3. Section 2 lacks any information on experimental replication. This is particularly worrisome. Please revise the manuscript detailing your experimental and technical replications (refering 10.1021/acsami.1c25014).
4. The Conclusion section should be strengthened: the important results and main conclusions drawn in this paper should be highlighted and presented in more precise language.
5. Where are these hydrogels going to be used in real life? Advantages of the designed CS-Tyramine-Gelatin hydrogels can be improved by comparing and citing 10.1016/j.carbpol.2017.10.088. It could be better if a brief comment (challenges and future prospects: 10.1016/j.reactfunctpolym.2020.104501) is added at the manuscript.
6. It is necessary to study the microstructure of hydrogels by SEM imaging of lyophilized hydrogels. The pore size of the gel is critical for cell culture.
Author Response
Reviewer 1
- There are some formatting errors in the article. For example, Line 20 and Line 81, Gelatin not Gelatine. Spelling of references must be checked to meet the journal style (such as Reference 15). Please check carefully and use abbreviation properly.
Response. Thank you for the remark. However, both terms mean the same and can be equally used depending on what style USA or England is used (https://mmingredients.co.uk/gelatine-or-gelatin/). We have corrected the manuscript using style “gelatin”. The style of the references, as well as abbreviations, were also checked. Some inaccuracies might happen due to the computer program adding the references.
- Figure 8, gelatin itself also has certain tyramine residues. Therefore, is it a combination of complexed gelatin and complexed CS when the HRP enzyme is cross-linked?
Response. Yes, you are right, it is a complex of CS, which was firstly synthesized with tyramine (see section 4.3) and then mixed with gelatin followed by HRP and H2O2 enzymatic cross-linking. We have introduced this information into the method section of the manuscript.
- Section 2 lacks any information on experimental replication. This is particularly worrisome. Please revise the manuscript detailing your experimental and technical replications (refering 10.1021/acsami.1c25014).
Response. Sorry for the missed information. The information about the statistical replicates was added to the section 2.
- The Conclusion section should be strengthened: the important results and main conclusions drawn in this paper should be highlighted and presented in more precise language.
Response. Thank you for the suggestion. The conclusion and result parts were corrected and strengthened.
- Where are these hydrogels going to be used in real life? Advantages of the designed CS-Tyramine-Gelatin hydrogels can be improved by comparing and citing 10.1016/j.carbpol.2017.10.088. It could be better if a brief comment (challenges and future prospects: 10.1016/j.reactfunctpolym.2020.104501) is added at the manuscript.
Response. Thank you for the suggestion. CS-Tyramine-Gelatin hydrogels are injectable type of hydrogels and definitely could be successfully used in the future to deliver various types of the cells and compounds into damaged articular joint. However, the efficiency of these applications still needs an additional proof not only in vitro, but also in vivo ranging from the model systems to human applications.
The synthesis of CS-Tyr was done according to a previous procedure with some modifications (Zhang, 2019). The reference and more detail description of hydrogel polymerization and evaluation was added to the Methods section. Gelation of CS-Tyr/gelatin was mediated via di-phenolic linkage using horseradish peroxidase (HRP) enzyme and hydrogen peroxide (H2O2). This polymerization mechanism has been used for various types of hydrogels. Because gelatin contains tyrosine residues, it can be crosslinked with CS-Tyr and form CS-Tyr/gelatin hydrogel. We did not find any previous study that fabricates chondroitin sulfate/gelatin hydrogel using this approach.
The swelling ratio of CS-Tyr/gelatin hydrogel based on its dried weight was about 30~35 folds. The elasticity of the hydrogels was analyzed using dynamic mechanical analysis. The additional information about the hydrogel composition, polymerization, elasticity and incorporation of the cells into this hydrogel is added to the method part.
Reference: Zhang, Y.J., et al., Injectable hydrogels from enzyme-catalyzed crosslinking as BMSCs-laden scaffold for bone repair and regeneration. Materials Science & Engineering C-Materials for Biological Applications, 2019. 96: p. 841-849.
Moreover, the possible future applications of CS-Tyramine-Gelatin hydrogels were described in added “Future prospects” section.
- It is necessary to study the microstructure of hydrogels by SEM imaging of lyophilized hydrogels. The pore size of the gel is critical for cell culture.
Response. Thank you for the suggestion. The pore size and the surface structure is critical for the various types of scaffolds rather than for the hydrogels. In this study, we have used injectable type of hydrogel, with the embedded/polymerized cells inside it. If the hydrogel would not be properly polymerized, the cells would escape from the hydrogel. On the other hand, if the hydrogel would be too tight, the cells would die. The cell viability test shown in this study confirmed that the cells polymerized in the hydrogel were alive after the long-term incubation in medium, which suggest the proper hydrogel-cell polymerization conditions allowing to penetrate nutrition necessary for the cell surviving.

Reviewer 2 Report
Uzieliene et al. present a manuscript evaluating the biocompatibility and chondrogenic differentiation of CS-Tyr/Gel hydrogels under mild mechanical load as well as the effect of the mechanical load on osteoarthritic human cartilage explants and their response to the CS-Tyr/gel/BM-BMSCs composite (in terms of the release of COMP and GAGs. The authors found that mechanical load induces the chondrogenic differentiation of BM-MSCs on the CS-Tyr/Gel hydrogel (based on the immunohistochemical detection of Collagen II); however, the same intensity of mechanical load induces a negative effect on the cartilage explants (determined as the release of COMP and GAGs). In addition, the co-incubation of CS-Tyr/Gel/BM-MSCs composite with the osteoarthritic human cartilage explants decreased the release of COMP and GAGs from the cartilage explants. This is an interesting manuscript. However, I have some comments:
I suggest editing English; some phrases need to be clarified, and easier to follow the message. For example, in line 39, “This limitation, together various types of mechanical joint damages,” in Line 435, “ … in the laboratory laminar ”. Line 437 “Using a biopsy needle, the cartilage was cut…”. These are some examples.
The authors did evaluate chondrocyte viability by measuring the release of LDH; however, there are protocols to determine chondrocyte viability, such as Syto 13, Calcein AM, and propidium iodide or other molecules; please indicate the reasons for selecting the LDH method.
Please indicate how BM-MSCs were characterized (using flow cytometry) in methods.
Concerning the synthesis of CS-Tyr, please indicate if the method was developed by the authors or is based on methods published in the literature. I also suggest including the chemical characterization of the CS-Tyr conjugate.
Please give more details concerning the chondroitin sulfate-tyramine hydrogel preparation; it needs to be clarified. Also, it is essential to provide information on how the BM-MSCs were encapsulated into CS-Tyr/Gel hydrogels.
For the detection of COMP, the supernatants of cartilage explants were diluted by 500 folds. Why was this dilution factor used?, did this dilution affect the detection of COMP?
Lines 131-138. Please move this information to the methods section.
Please improve the resolution of the figures.
In lines 134, 497, and 560, I do not understand the meaning of “250k”.
The immunohistochemical detection of collagen type II was performed to evaluate chondrogenic differentiation. Molecules such as collagen X and aggrecan are also crucial as chondrogenic markers. Did the authors evaluate these molecules?. Also, did the authors consider those genes’ gene expression (ACAN or COL2A1)?, if not, it is important to indicate that this could be a limitation of the study.
Author Response
Reviewer 2
Uzieliene et al. present a manuscript evaluating the biocompatibility and chondrogenic differentiation of CS-Tyr/Gel hydrogels under mild mechanical load as well as the effect of the mechanical load on osteoarthritic human cartilage explants and their response to the CS-Tyr/gel/BM-BMSCs composite (in terms of the release of COMP and GAGs. The authors found that mechanical load induces the chondrogenic differentiation of BM-MSCs on the CS-Tyr/Gel hydrogel (based on the immunohistochemical detection of Collagen II); however, the same intensity of mechanical load induces a negative effect on the cartilage explants (determined as the release of COMP and GAGs). In addition, the co-incubation of CS-Tyr/Gel/BM-MSCs composite with the osteoarthritic human cartilage explants decreased the release of COMP and GAGs from the cartilage explants. This is an interesting manuscript. However, I have some comments:
I suggest editing English; some phrases need to be clarified, and easier to follow the message. For example, in line 39, “This limitation, together various types of mechanical joint damages,” in Line 435, “ … in the laboratory laminar ”. Line 437 “Using a biopsy needle, the cartilage was cut…”. These are some examples.
Response. Thank you for the suggestion. The English language was checked and corrected.
Question. The authors did evaluate chondrocyte viability by measuring the release of LDH; however, there are protocols to determine chondrocyte viability, such as Syto 13, Calcein AM, and propidium iodide or other molecules; please indicate the reasons for selecting the LDH method.
Response. Thank you for the question. There are various methods to measure cell viability. However, not all of them equally suits to measure cells in 2D and 3D ways. The chondrocyte viability in cartilage explants also requires 3D cell viability methods, i.e. the cell viability in 3D hydrogels and/or cartilage explants is better to measure by component release-based methods like LDH that allow to perform a quantitative evaluation. The intracellular metabolic methods like Calcein AM and other better suits to measure cell viability in 2D way than in 3D way. However, we also checked cell viability in hydrogels using Calcein AM, which is presented in (Figure 3F) (Live/Dead cell viability measurement).
Question. Please indicate how BM-MSCs were characterized (using flow cytometry) in methods.
Response. Sorry for the missed information. The flow cytometry measurement method was added to the method part.
Question. Concerning the synthesis of CS-Tyr, please indicate if the method was developed by the authors or is based on methods published in the literature. I also suggest including the chemical characterization of the CS-Tyr conjugate.
Response. We thank for the comment. The synthesis of CS-Tyr was done according to a previous procedure (1). Recently, our manuscript with the CS-Tyr/Gel and BMMSCs and chondrocytes was accepted in IJMS, but still does not have a bibliographic data. The reference was added to the Methods section.
Gelation of CS-Tyr/gelatin was mediated via di-phenolic linkage using horseradish peroxidase (HRP) enzyme and hydrogen peroxide (H2O2) (1). This mechanism was used for various types of hydrogels. Because gelatin contains tyrosine residues, it can be crosslinked with CS-Tyr and form CS-Tyr/gelatin hydrogel. We did not find any previous study that fabricates chondroitin sulfate/gelatin hydrogel using this approach beside the recently accepted our manuscript with the CS-Tyr/Gel and BMMSCs and chondrocytes in IJMS, which but still does not have a bibliographic data.
The swelling ratio of CS-Tyr/gelatin hydrogel based on its dried weight was about 30~35 folds. The elasticity of the hydrogels was analyzed using dynamic mechanical analysis. The additional information about the hydrogel composition, polymerization, elasticity and incorporation of the cells into the hydrogel is added to the method part.
- Zhang, Y.J., et al., Injectable hydrogels from enzyme-catalyzed crosslinking as BMSCs-laden scaffold for bone repair and regeneration. Materials Science & Engineering C-Materials for Biological Applications, 2019. 96: p. 841-849.
Question. Please give more details concerning the chondroitin sulfate-tyramine hydrogel preparation; it needs to be clarified. Also, it is essential to provide information on how the BM-MSCs were encapsulated into CS-Tyr/Gel hydrogels.
Response. Sorry for the missed information. More detail description of hydrogel preparation and cell encapsulation is added to the method part 4.4.
Question. For the detection of COMP, the supernatants of cartilage explants were diluted by 500 folds. Why was this dilution factor used?, did this dilution affect the detection of COMP?
Response. The cartilage explants have very high level of COMP, which is impossible to detect by ELISA in undiluted form. The 500-fold dilution was experimentally optimized for the used ELISA kit.
Question. Lines 131-138. Please move this information to the methods section.
Response. The mentioned lines were moved to the Method section.
Question. Please improve the resolution of the figures.
Response. The resolution worsens when manuscript is transferred to the pdf format. The journal has all Figures in png format and we have improved quality of the images.
Question. In lines 134, 497, and 560, I do not understand the meaning of “250k”.
Response. The “250k” is a shorter way to write “250 thousand” of the cells. The abbreviation is changed.
Question. The immunohistochemical detection of collagen type II was performed to evaluate chondrogenic differentiation. Molecules such as collagen X and aggrecan are also crucial as chondrogenic markers. Did the authors evaluate these molecules?. Also, did the authors consider those genes’ gene expression (ACAN or COL2A1)?, if not, it is important to indicate that this could be a limitation of the study.
Response. Thank you for the question. Yes, we agree that ECM components like aggrecan and other are important for the chondrogenic differentiation. However, so far it is agreed that the collagen II is the most important.
In addition, some studies suggest to be careful with aggrecan and collagen X, as markers of chondrogenesis and chondrocyte hypertrophy, respectively, since their expression is not consistent particularly during the MSCs differentiation process (Fackson Mwale 1 , Dorothy Stachura, Peter Roughley, John Antoniou Limitations of using aggrecan and type X collagen as markers of chondrogenesis in mesenchymal stem cell differentiation. J Orthop Res. 2006 Aug;24(8):1791-8. doi: 10.1002/jor.20200.). The study suggests that aggrecan and type X collagen, as a chondrogenic differentiation biomarkers, should be used with precautions since might not always reflect a real situation.
The PCR measurements usually require higher number of hydrogel/cell composites due to the complicated isolation of the RNR from the hydrogels. However, it is well known that gene expression not always correspond the protein level, particularly in cartilage tissue with high level of ECM. The change of ECM components inside the OA cartilage and outside during the mechanical load is much more informative than gene expression, at least in this study.

Author Response
We thank to Reviewer for the valuable and kind remarks, questions and suggestions that hopefully improved the manuscript.
Reviewer 3
Manuscript ID: ijms-2164214
Title: Chondroitin Sulfate-Tyramine-based Hydrogels for Cartilage Tissue Repair
Authors: Ilona Uzieliene, Daiva Bironaite1, Jolita Pachaleva, Edvardas Bagdonas, Arkady Sobolev, Wei-Bor Tsai, Giedrius Kvederas, Eiva Bernotiene.
In the paper entitled: “Chondroitin Sulfate-Tyramine-based Hydrogels for Cartilage Tissue Repair” the authors reported the use of chodroitin sulfate-tyramie-gelatin hydrogels loaded with bone marrow mesenchymal stem cells for the treatment and regeneration of osteoarthritis cartilage. The designed hydrogels were incubated with damaged cartilage explants in vitro under mechanical loading, observing a remarkable decrease on the release of some markers related with cartilage degeneration.
Overall, I found the paper suitable for publication in International Journal of Molecular Sciences after major revisions. I consider the results section acceptably detailed and reasonably organized, however some experiments should be explained with more detail and the quality/resolution of the figures must be also improved. In the Introduction part, I suggest improving English style and providing more related bibliography to properly address the state of the art and the significance of the proposed work. Following, I expose some comments and suggestions that could upgrade the paper and which I would like the authors to address before consider resubmission.
FORMATTING AND STYLE
Question. The quality of the images could be notably improved. Captions in Figures 1, 2, 3 and 4 are rather hard to read.
Response. Thank you for the suggestion. The worse images quality became as the manuscript was converted to the PDF. The original high quality images were sent to the journal.
Question. Figure 1: label “A” missing
Response. The “A” was hid behind the Figure. It is corrected now.
Question. Size of the scales is very small in all Figures.
Response. The scale bars in the figures are originally made by taking an image and cannot be enlarged due to the original Figure size. The figures are provided in higher resolution.
ABSTRACT
Question. The work performed by the authors is presented in a quite detailed manner. However, when referring to the mechanical effects, the expression used is “mild mechanical load” which can be too loose and does not describe precisely enough the procedure. It could be good to elaborate further and explain in greater depth the employed technique.
Response. Thank you for the comment. First of all, I would like to explain the understanding of “mild mechanical load” and “overload”. There are various types of mechanical compression systems in vitro mimicking loads that cartilage suffers during human walk or intensive sport and they cannot be compare to each other, i.e. the parameters of mechanical load in vitro strongly depend on many factors, i.e. used compression equipments, hydrogels, cells, duration of compression, size and origin of cartilage explants and many other experimental conditions. Therefore, it is impossible to compare all the mechanical compression systems used in vitro with the each other or with the natural cartilage compression in vivo.
The mechanical compression used in this study for the hydrogels (10kPa, 1Hz, 1h/day for 7 days) positively affected encapsulated cells, i.e. increased the level of collagen II and could be named as a mild mechanical load, while compression used for the OA cartilage explants (30kPa, 1Hz, 1h/day for 7 days) decreased intracellular level of aggrecan and stimulated release of ECM components and could be named as a strong one or overload. Therefore, in this study we wanted to show, can the hydrogel/cell composite protect OA cartilage explants in vitro under strong mechanical load negatively affecting cartilage.
Previously we have reviewed the mechanotransductive properties of different biomimetic hydrogels and ECM components-based systems used for the cartilage tissue engineering and chondrogenic differentiation in vitro (Uzieliene et al., 2021). It is known that under physiological conditions, compressive modulus of articular cartilage varies from 0.4–2.0 MPa (Lee et al., Biomechanics of Cartilage and Osteoarthritis, 2015), which is hardly achievable in vitro.
So, if the mechanical load worsens functioning of the chosen experimental objects (hydrogels/cells, explants and other) evaluated at ECM, intracellular protein or gene levels it can be named as a mechanical overload, while a load positively affecting cells and/or cartilages – as a mild one. Following the reviewer’s suggestion, we have added such an explanation to the introduction and discussion parts.
Question. GAG abbreviation is defined twice in the abstract (Line 18 and Line 25).
Response. One abbreviation was used for a singular form, another – for a plural. We left just a plural form.
INTRODUCTION
Overall, typographical errors and English style should be revised all throughout the document. For instance, in the introduction part alone:
Line 57 - instead of: “are most widely used”, should be: “are the most widely used”.
Line 66 - instead of: “have”, should be: “has”.
Line 71 - instead of: “is important factor” should be: “is an important factor”.
(...)
Response. Sorry for the left typing mistakes. The manuscript has been edited by an English-speaking native.
Question. Avoid the definition of abbreviations only used once during the manuscript (e.g., FDA in line 50).
Response. All abbreviations are checked.
Question. To properly address the state of the art I suggest at least mentioning other potential treatments of damaged cartilage and other tissues besides cell delivery using polymeric hydrogels, such as tissue engineering approaches. You can also briefly refer some recently developed techniques with impressive potential to design hydrogels with the adequate characteristics for this kind of applications such as layer by layer technique or 3D bioprinting (see https://doi.org/10.1021/acs.cgd.9b00831 https://doi.org/10.1039/D1TB00717C or https://doi.org/10.1002/adfm.202208940)
Response. Thank you for the suggestion. The future possible hydrogel strategies and their applications were added to the discussion part including suggested citations.
Question. You must explain with much more detail why you chose chondroitin sulfate-tyramie-gelatin formulation. Why these compounds? Why a mixture of them?
Response. We thank for the comment. Our co-authors from Taiwan are specializing in formation of enzymatically crosslinked hydrogels for the articular joint regeneration. Since the mechanical load plays very important role in OA cartilage damage or regeneration, we raised the hypothesis that the application of this hydrogel comprising natural components of cartilage ECM with the encapsulated BM-MSCs could be beneficial for the protection of OA cartilage explant under the mechanical load in vitro.
The synthesis of CS-Tyr was done according to a previous procedure with some modifications (Zhang, 2019). The reference has been added to the Methods section as well as hydrogel polymerization and cell encapsulation was also described in more details.
Gelation of CS-Tyr/gelatin was mediated via di-phenolic linkage using horseradish peroxidase (HRP) enzyme and hydrogen peroxide (H2O2) (Zhang, 2019). This mechanism in not new, it has been used for various types of hydrogels, but because gelatin contains tyrosine residues, it can be crosslinked with CS-Tyr and form CS-Tyr/gelatin hydrogel. We did not find any previous study that fabricates chondroitin sulfate/gelatin hydrogel using this approach.
In addition, CS-Tyr/gelatin hydrogel properly traps and carries cells, is biocompatible during a long-term of chondrogenic differentiation, which is vitally important for the cartilage regeneration particularly under the mechanical load. Moreover, the CS-Tyr/gelatin hydrogel, beside the cells, might carry various types of bioactive compounds positively affecting cartilage tissue.
The more information about the alternative hydrogels was added to the manuscript.
Reference: Zhang, Y.J., et al., Injectable hydrogels from enzyme-catalyzed crosslinking as BMSCs-laden scaffold for bone repair and regeneration. Materials Science & Engineering C-Materials for Biological Applications, 2019. 96: p. 841-849.
RESULTS
Question. I suggest rebuilding this figure 1, in the current version the position of “chondrogenic dif.” is confusing.
Response. The Figure 1 was rebuilt; we believe it became less confusing.
Question. Can you add quantitative evaluation of chondrogenic differentiation in Figure 1B?
Response. The level of ECM component COMP during the chondrogenic differentiation of BM-MSCs was added.
Question. Line 134: express the number of cells in units per mL would be better in my opinion.
Response. Unfortunately, it is impossible. The units are usually used to express activity of enzymes or antibiotics. In this assay, the dyes used to identify differentiated cells, stain proteins that appear during the differentiation and then are extracted from the cells. So, the absorption difference between the differentiated and control (not differentiated) cells is the best way to show the intensity of differentiation.
Question. Figure 2A: The image corresponds to four equal samples? In this case I suggest showing only one; or clarify that samples are equal in the legend.
Response. Thank you very much for your suggestion. The statement “The representative micrographs of hydrogel and histological staining are shown.” was missing and is added now.
Question. Line 175: I suggest incorporating a brief description of the characteristic of the applied mechanical load here in results section (frequency, intensity…)
Response. Thank very much for the suggestion. However, in the Fig. 2 there was no mechanical load; it is a biocompatibility test. The explanation about the mechanical load is added to the Fig. 3.
Question. Line 182: I think that detail here the use of calcein and PI for live/dead staining is not necessary. This info corresponds to methods section.
Response. Thank you for the suggestion. We think that very short explanation will help to the readers to better understand which color means what.
Question. Figure 7: COMP release after 21 days is slightly higher in medium when it is supplemented with TGF-β3, but GAGs release is lower. Can you hypothesize about the reason?
Response. Thank you for the interesting question. The explanation, could be that TGF-β3 is not equally quickly stabilize or increase synthesis of COMP and GAGs. The stabilization of COMP might need a higher TGF-β3 concentration and/or longer incubation time in order to better decrease its release from the OA cartilage explants.
DISCUSSION
Line 348: You should also mention that the complex anisotropic architecture of cartilages hinders their treatment and the development of adequate regeneration strategies (you can see the last section of this recent review article where the anisotropic structure and composition of cartilage tissues is described: https://doi.org/10.1021/acsnano.0c08253).
Response. The description of cartilage treatment due to the anisotropic structure of cartilage is added to the introduction part.
Question. I suggest moving the last part of this discussion section to conclusions. It makes more sense there and provides the conclusions section with more robustness than in the present form.
Response. Thank you for the valuable suggestion. It was moved to the conclusions.
MATERIALS and METHODS
Question. “Lines 519 to 522”. Statement from “The resulting…” to “… vigorous stirring.” is repeated.
Response. The repeated sentence was deleted.
Question. The source of the Hydroxyapatite crystals is not detailed in the text.
Response. Calcium crystals that are produced by BMMSCs undergoing osteogenic differentiation are precursors of hydroxyapatite, so it is better to write calcium crystals and they characterizing osteogenic lineage, usually stained with Alizarin red for proof and quantification.
Question. “Line 440”. Inconsistent use of quotation marks. This typographic error is repeated all throughout the paper.
Response. The quotation style depends on the journal’s template, which is adding such type of quotations.

Round 2
Reviewer 3 Report
I would like to thank the authors for conscientiously addressing all the comments raised in the first revision of the manuscript. The new version can be published in its current form in International Journal of Molecular Sciences.